# External validation of EPIC's Risk of Unplanned Readmission model, the LACE+ index and SQLape as predictors of unplanned hospital readmissions: A monocentric, retrospective, diagnostic cohort study in Switzerland

**Aljoscha Benjamin Hwang**[1,2]*, **Guido Schuepfer**[1]☺, **Mario Pietrini**[1]☺, **Stefan Boes**[2]☺

**1** Staff Medicine, Cantonal Hospital Lucerne, Lucerne, Switzerland, **2** Department of Health Sciences and Medicine, University of Lucerne, Lucerne, Switzerland

☺ These authors contributed equally to this work.
* aljoscha.b.hwang@hotmail.com

## Abstract

### Introduction

Readmissions after an acute care hospitalization are relatively common, costly to the health care system, and are associated with significant burden for patients. As one way to reduce costs and simultaneously improve quality of care, hospital readmissions receive increasing interest from policy makers. It is only relatively recently that strategies were developed with the specific aim of reducing unplanned readmissions using prediction models to identify patients at risk. EPIC's Risk of Unplanned Readmission model promises superior performance. However, it has only been validated for the US setting. Therefore, the main objective of this study is to externally validate the EPIC's Risk of Unplanned Readmission model and to compare it to the internationally, widely used LACE+ index, and the SQLAPE® tool, a Swiss national quality of care indicator.

### Methods

A monocentric, retrospective, diagnostic cohort study was conducted. The study included inpatients, who were discharged between the 1st of January 2018 and the 31st of December 2019 from the Lucerne Cantonal Hospital, a tertiary-care provider in Central Switzerland. The study endpoint was an unplanned 30-day readmission. Models were replicated using the original intercept and beta coefficients as reported. Otherwise, score generator provided by the developers were used. For external validation, discrimination of the scores under investigation were assessed by calculating the area under the receiver operating characteristics curves (AUC). Calibration was assessed with the Hosmer-Lemeshow $X^2$ goodness-of-fit test This report adheres to the TRIPOD statement for reporting of prediction models.

**Data Availability Statement:** A minimal anonymized data set necessary to replicate the study findings was uploaded to the platform DRYAD (https://doi.org/10.5061/dryad. 70rxwdbxw).

**Funding:** The author(s) received no specific funding for this work.

**Competing interests:** The authors have declared that no competing interests exist.

## Results

At least 23,116 records were included. For discrimination, the EPIC´s prediction model, the LACE+ index and the SQLape® had AUCs of 0.692 (95% CI 0.676–0.708), 0.703 (95% CI 0.687–0.719) and 0.705 (95% CI 0.690–0.720). The Hosmer-Lemeshow $X^2$ tests had values of p<0.001.

## Conclusion

In summary, the EPIC´s model showed less favorable performance than its comparators. It may be assumed with caution that the EPIC´s model complexity has hampered its wide generalizability—model updating is warranted.

## Introduction

### Background

Readmissions after acute care hospitalization are relatively common, costly to the health care system and associated with a significant burden for patients [1–5]. A readmission increases the risk of dependence and functional or psychosocial decline [5]. Moreover, readmission increases the risk of decompensation of other comorbid conditions, thus increasing the frailty of elderly patients [6].

The belief that readmission rates are a valid indicator to assess quality of care has led to their inclusion in hospital quality surveillance [6, 7]. In December 2020, the Swiss National Association for Quality Development in Hospitals and Clinics (ANQ) reported its most recent findings. Accordingly, based on 2018 figures, the number of Swiss hospitals that reported more readmissions, as expected according to their patient mix, declined from a high in 2016. In total, 26 out of 193 hospitals reported rates (observed/expected) outside the norm, i.e., significantly higher than 1 [8].

In the last few years, the observed increase in costs has posed major challenges for the healthcare system. Healthcare costs in Switzerland have risen by a third within the last decade [9]. As one way to reduce costs and simultaneously improve quality of care, unplanned hospital readmissions have received increasing interest from policy makers. It is only relatively recently that policies were developed with the specific aim of reducing unplanned readmissions. In Switzerland, the readmission policy involves financial penalties, i.e., that patient records of the first admission and the relevant readmission are merged into a single case if certain criteria are met. Consequently, hospitals receive only one DRG-based payment for both admissions [10]. As a result, although some readmissions cannot be avoided and the proportion of potentially avoidable readmissions (PARAs) remains debatable, health care organizations invest considerable resources in efforts to reduce unplanned hospital readmissions [11–13].

To most efficiently reduce unplanned readmissions, hospitals need to target effective discharge and post-discharge interventions at those who need them the most. One of the more recent strategies is the application of prediction models. As systematic reviews have shown, there are many models aimed at identifying those at greater risk of readmission [14, 15]. The majority include readily available predictors such as demographic and administrative data, or even comorbidities, laboratory results, and medications [14]. Among these models, the Epic Risk of Unplanned Readmission model, developed in 2015 for the U.S. acute care hospital setting, promises superior calibration and discriminatory abilities. The model was developed by

Epic Systems Corporation based on data from 26 Epic community member hospitals, including more than 275,000 inpatient hospital admission encounters, to determine a patient's risk of unplanned readmission within 30 days of being discharged from an index admission.

## Rationale

Rising awareness about the importance of electronic health records (EHRs) for enhancing the efficiency and quality of patient care has augmented global electronic health records industry growth. In fall 2019, the Lucerne Cantonal Hospital rolled out Epic's EHR system as the first hospital in a German-speaking country. Herewith, the conditions to apply more complex (in terms of the included number and type of predictors) prediction models were created. With the intention of routine application, the Epic Risk of Unplanned Readmission model was externally validated. Although the Epic model was developed for the acute care hospital setting, variations in demographic features, disease prevalence, and differences in test conditions (e.g., defining criteria of relevant readmissions, time frame of measurement) entailed external validation prior to routine application in the Swiss acute care hospital setting. External validation means applying the model with its predictors and assigned weights, as estimated from the development study, to a new population; measuring the predictor and outcome values; and quantifying the model's predictive performance (calibration and discrimination) [16].

For comparison, the SQLape® tool (Striving for Quality Level and Analyzing of Patient Expenditures), a Swiss national quality of care indicator that has become a quasi-standard in Switzerland and allows the prediction of hospital readmissions, was included [17]. Based on a systematic review of models to predict unplanned hospital readmissions from 2016 [14] and a literature search on PubMed for validation studies published after 2015, the LACE+ model was included as the second comparative model. The LACE+ score is easily producible and has been analyzed in various prospective and retrospective studies, including studies with medical inpatient cohorts from Swiss tertiary care providers [18–22].

## Objective

The main objective of this study is to externally validate the Epic Risk of Unplanned Readmission model as a predictor of unplanned hospital readmissions within 30 days and to compare its predictive ability with that of the LACE+ index and the SQLape® readmission algorithm.

## Methods

### Design

This monocentric, retrospective, diagnostic cohort study included inpatient hospitalization cases from the Lucerne Cantonal Hospital (LUKS), which is the largest tertiary healthcare provider in Central Switzerland with a beneficiary population of ~ 800,000. The LUKS is a three site, 800-bed hospital with all medical and surgical specialties present, four Level 3 intensive care units, and four 24 h/7 days per week emergency departments (EDs). This study was approved by the Ethics Committee Northwest- & Central Switzerland (October 7, 2019, project-ID 2019–01861). An informed consent was not obtained, because this study was conducted as a quality control project that used anonymized data. This study was conducted according to the principles of the Declaration of Helsinki.

### Participants

All inpatients between one and 100 years old who were discharged between 1 January 2018 and 31 December 2018 were included as Cohort A. Inpatients discharged between 23 of

September 2019 and 31 December 2019 were included as Cohort B. Inpatients were excluded as follows: (a) admissions/transfers from another psychiatric, rehabilitative, or acute care ward from the same institution; (b) discharge destinations other than the patient's home, considered treatment continuation; (c) foreign or unknown residence; and (d) deceased before discharge. For individuals with multiple hospitalizations, only the first hospital stay was included in the analysis.

## Outcome

The study outcome was unplanned 30-day readmission to the same hospital. An unplanned readmission was defined as an urgent readmission, i.e., not scheduled in advance and requiring treatment within 12 hours [23]. No more than one readmission for each discharge was considered.

## Prediction models

The following paragraph provides a brief description of the prediction models evaluated in this study. It should be noted that the Epic Risk of Unplanned Readmission and the SQLape® model are commercially distributed products. Implicitly, due to copyright issues, not all information about the prediction models required to replicate this validation study was disclosed in sufficient detail. Replicating this study requires licensing.

Epic Risk of Unplanned Readmission model: The Epic Risk of Unplanned Readmission model is a logistic regression model that predicts the risk of unplanned readmissions within 30 days of the index hospital discharge date. An unplanned readmission was defined by the Centers for Medicare & Medicaid Services (CMS) in the 2015 Measure Information About the 30-Day All-Cause Hospital Readmission Measure, Calculated For the Value-Based Payment Modifier Program [24]. Adaptations were made and included patients aged between 1 and 100 years at the time of admission, patients of any payer, and hospital encounters for which patients left against medical advice. The development data set included more than 275,000 hospital inpatient encounters from 26 different hospitals. Three of these hospitals were large academic medical centers (1000+ beds each), while the others were either smaller regional or community hospitals. All included sites were chosen from very distinct geographic regions in the US to ensure as diverse a population as possible. Selection by specialties/medical disciplines was not applied.

After feature selection, using a least shrinkage and selection operator (LASSO) penalty, the final model consisted of 27 predictive parameters [25]. The internal and external validation of the model showed acceptable discrimination at predicting unplanned readmission within 30 days post discharge, with an area under the receiver operating characteristics curve (AUC of the ROC curve) ranging from 0.69 to 0.74 [26].

LACE+: The LACE+ risk index is a point score derived from a logistic regression model that was developed to predict the risk of 30-day postdischarge death or urgent readmission [27, 28]. It was developed and internally validated based on a large, randomly selected, population-based sample from Ontario, Canada in 2012. The development sample excluded patients who underwent same-day surgeries and psychiatric and obstetric admissions. Backward feature selection was performed (with a significance level of $\alpha = 0.05$) and resulted in 11 significant parameters [29]. The final point score ranges from -15 to 114, and a score greater than 90 is considered to indicate a high risk for urgent readmission or death within 30 days after discharge. The internal validation of the 11-item index, excluding the Canada-specific case-mix group (CMG) score, showed acceptable discrimination with an AUC of 0.743 for urgent readmission only but poor calibration (H-L statistic 58.93, $p < 0.0001$) [27].

SQLape®: The SQLape® model (Striving for Quality Level and Analyzing of Patient Expenditures), a computerized validated algorithm, was developed in 2002 in Switzerland [6]. The SQLape® model predicts potentially avoidable hospital readmissions within 30 days after hospital discharge. An unplanned readmission was defined according to the "Algorithm for the Identification of Potentially Avoidable Rehospitalizations" [8]. The development sample consisted of 131,809 inpatient stays from 49 Swiss acute care hospitals (including the Lucerne Cantonal Hospital), of which 12 hospitals were located in the French-speaking part of Switzerland. Among others, healthy newborns, residents outside of Switzerland, and elective surgical patients who could usually receive same-day surgery were excluded. After backward elimination was performed, the Poisson regression model consisted of six variable groups. Of the 131,809 inpatient stays mentioned above, 66,069 were used for internal validation. Discrimination was measured by Harrell's C statistic, which is also referred to as the estimated area under the receiver operating characteristics (ROC) curve (AUC). A value of 0.72 showed acceptable discriminative ability [6].

## Descriptive and predictive variables

The following data were retrospectively extracted from an enterprise data repository that integrates routinely collected information from multiple clinical information (CIS) and enterprise resource planning systems (ERPs):

- Socio-demographic data: Date of birth, gender, nationality, postal code, type of medical insurance

- Hospital administrative data: Patient origin (home, nursing home, or other institution), admission date, length of stay (LOS), admission type (elective, urgent, etc.), discharge date, discharge destination (home, nursing home, or other institution), discharge decision (initiated by the physician, initiated by the patient, etc.), cost weight, diagnostic-related group (DRG), primary diagnosis, procedure codes, readmission date, readmission reason, final discharge date, major diagnostic category, admission and discharge medical specialty, and admission and discharge ward

- Clinical data: Charlson Comorbidity Index (CCI), imaging orders, electrocardiogram, specific laboratory results, and medications (clinical data were only extracted for Cohort A)

- Risk of Unplanned Readmission score: 8 a.m. and 12 a.m. scores (scores were only extracted for Cohort B)

All prediction model input parameters (predictors) are detailed in Table 1 (Model Predictors). While the SQLape® model was developed within the Swiss context, the LACE+ and Epic models were designed for and trained on patient populations outside of Switzerland. For this reason, aspiring model validation and implementation considerations were followed by the adaptation of certain model input parameters to "alleviate" setting specific discrepancies. In regard to the LACE+ model, only minor adaptations were carried out. First, the case-mix group (CMG) score was excluded because CMG scores can only be calculated for hospital admissions inside Canada (CMGs aggregate acute care inpatients with similar clinical and resource-utilization characteristics) [27]. Second, all codes from the International Classification of Diseases, 10th Revision, Clinical Modification (ICD-10-CM) used by the Charlson Comorbidity Index (CCI) [30] to quantify patient burden of disease were mapped onto ICD-10 codes of the German Modification (GM) version, which is used in Switzerland. This was done by a clinical expert with extensive working experience in medical coding. Epic's Risk of Unplanned Readmission model required much more comprehensive adjustments due to its

**Table 1. Model predictors.**

| Model | Data category | Variable | Variable type | Unit / categories | Working definition | Adaptations | Differences Cohort A | Cohort B | Data points per observation |
|---|---|---|---|---|---|---|---|---|---|
| Epic Risk of Unplanned Readmission model | Demographics | Age | Numeric | Years | The age at the day of hospital admission. | - | - | - | Multiple |
| | Administrative data | Current length of hospital stay | Numeric | | The number of days of hospitalization of the ongoing stay, from the time of inpatient admission till the point in time of score calculation. It does consider the time spent in the Emergency department (ED)—rounded to 3 decimal points (Time stamp at model calculation minus hospital admission time stamp). | - | X | - | Multiple |
| | Resource utilization | Number of past ED visits in the last 6 months | Numeric | Visits | A count of the number of ED visits in the last six months. Counts both, those where the patient went home healthy, and the ones, where the patient was subsequently submitted to the wards. The look-back period starts at the day of admission. | - | - | - | Multiple |
| | | Number of past admissions in the last 12 months | | Visits | Count of the number of inpatient stays in the last 12 months. It includes hospitalizations no matter how many days the patient has stayed (the patient does not need to stay for the night; admission and same day discharge stays are included). ED visits without transfer to the ward and ambulant office visits are excluded. The admission type (urgent, elective, etc.) is not relevant. The look-back period starts at the day of admission. | - | - | - | Multiple |
| | | Has future scheduled appointments | Categorical | Yes/No | Checks whether the patient has an outpatient appointment scheduled for any time after the day of the readmission risk score calculation? Planned hospital stays are not counted. There is no maximum look-forward period. Any scheduled appointment in the future is considered. | - | - | - | Multiple |

(*Continued*)

**Table 1.** (Continued)

| Model | Data category | Variable | Variable type | Unit / categories | Working definition | Adaptations | Differences Cohort A | Cohort B | Data points per observation |
|---|---|---|---|---|---|---|---|---|---|
| | | Prior length of stay of 10 days or more in the last 12 months | Categorical | Yes/No | Checks whether the patient had a hospital stay of at least 10 days (LOS) in the last 12 months. The look-back period starts at the day of admission. | - | - | - | Multiple |
| | Medications | Number of active medication orders | Numeric | Orders | Counts the total number of prescribed medications at the point in time of model calculation. Includes patient's medication on demand only if administered. Does not include entry/outpatient medication. Several prescriptions of the same medication with the same dosage count as one prescription/active medication; prescriptions of the same medication but as varying dosage on the same day count separately. Prescriptions of the same active ingredient but through various routes of administration (orally, intravenously, etc.) count as separate prescriptions; prescriptions of the same active ingredient but as different medicinal products count as separate prescriptions. | - | X | - | Multiple |
| | | Anticoagulants | Categorical | Yes/No | Checks whether the patient, at the time of risk score calculation, has active orders belonging to certain ATC groups. The ATC groups are detailed in S1 Appendix. | X | - | - | Multiple |
| | | Non-Steroidal Anti-Inflammatory drugs (NSAIDs) | Categorical | Yes/No | | X | - | - | Multiple |
| | | Corticosteroids | Categorical | Yes/No | | X | - | - | Multiple |
| | | Antipsychotics | Categorical | Yes/No | | X | - | - | Multiple |
| | | Ulcer medication | Categorical | Yes/No | | X | - | - | Multiple |
| | Comorbidities | Diagnosis of cancer | Categorical | Yes/No | Checks whether the patient has a diagnosis belonging to the corresponding ICD-10 GM grouper at day of discharge? A list of exact codes used to identify relevant disorders is available on reasonable request from the corresponding author. | X | X | - | Single |
| | | Diagnosis of deficiency anemia | Categorical | Yes/No | | X | X | - | Single |
| | | Diagnosis of electrolyte disorder | Categorical | Yes/No | | X | X | - | Single |
| | | Diagnosis of renal failure | Categorical | Yes/No | | X | X | - | Single |
| | | Diagnosis of drug abuse | Categorical | Yes/No | | X | X | - | Single |

(*Continued*)

**Table 1.** (Continued)

| Model | Data category | Variable | Variable type | Unit / categories | Working definition | Adaptations | Differences Cohort A | Cohort B | Data points per observation |
|---|---|---|---|---|---|---|---|---|---|
| | | Charlson Comorbidity Index (EPIC version) | Numeric | Points | To calculate the adapted Charlson Comorbidity Index (CCI) the following formula was used: The Charlson Comorbidity Index ranges between 0 and 32 points, and is based on the following diagnoses: 1 pt.—Myocardial Infarction; 1 pt.—Peripheral Vascular Disease; 1 pt.—Cerebrovascular Disease; 1 pt.—Diabetes w/o chronic complications; 2 pts.–Cancer; 2 pts.—Mild Liver Disease; 2 pts.—Chronic Pulmonary Disease; 2 pts.—Congestive Heart Failure; 3 pts.–Dementia; 3 pts.—Rheumatic Disease; 4 pts.–HIV/AIDS; 4 pts.—Moderate or Severe Liver Disease; 6 pts.—Metastatic Solid Tumor. The original groupers, based on ICD-10 CM codes, were replicated containing mapped ICD-10 codes according to the German modification (GM). The comorbidity score was calculated based on all known diagnoses at the day of discharge. A list of exact codes used to compute the Charlson Comorbidity Index (CCI) is available on reasonable request from the corresponding author. | X | X | - | Single |
| | Biological data | Hemoglobin value (g/dl) | Categorical | Normal/abnormal | Checks at the point in time of risk score calculation, whether the most recent lab test result of the last 72 hours was abnormal according to corresponding reference ranges. The exact lab components used to identify all relevant laboratory test results are detailed in S2 Appendix. | X | - | - | Multiple |
| | | Calcium value (mg/dl) | Categorical | Normal/abnormal | | X | - | - | Multiple |
| | | Blood Urea Nitrogen (BUN) value (mg/dl) | Categorical | Normal/abnormal | | X | - | - | Multiple |
| | | Creatinine value (mg/dl) | Categorical | Normal/abnormal | | X | - | - | Multiple |
| | | Prothrombin Time and International Normalized Ratio (PT/INR) value (ratio) | Categorical | Normal/abnormal | | X | - | - | Multiple |
| | | Phosphate tested | Categorical | Yes/No | Checks at the point in time of risk score calculation, whether the patient had a phosphate lab test done in the last 3 days? The look-back period starts at the point in time of risk score calculation. | X | - | - | Multiple |

(*Continued*)

**Table 1.** (Continued)

| Model | Data category | Variable | Variable type | Unit / categories | Working definition | Adaptations | Differences Cohort A | Differences Cohort B | Data points per observation |
|---|---|---|---|---|---|---|---|---|---|
| | Interventions/ orders | Imaging orders | Categorical | Yes/No | Checks at the point in time of risk score calculation, whether the hospital has provided an order of this type to the patient in the last six months? / Has the hospital documented any related "tarif medical" (TARMED) service codes of the TARMED chapter 39 (catalogue version 1.09, valid from 01.01.2018) as part of the entry of services rendered? | X | - | - | Multiple |
| | | Restraining orders | Categorical | Yes/No | Not relevant. | - | - | - | Multiple |
| | | Electrocardiography (ECG) | Categorical | Yes/No | Checks at the point in time of risk score calculation, whether the hospital has provided an order of this type to the patient in the last six months? / Has the hospital documented any related "tarif medical" (TARMED) service codes of the following (catalogue version 1.09, valid from 01.01.2018) as part of the entry of services rendered?:<br>• 17.0010 Electrocardiogram(ECG).<br>• 17.0080 Exercise ECG<br>• 17.0090 Exercise ECG, Ergometry<br>• 17.0120 ECG rhythm strip, per 5 minutes<br>• 17.0130 ECG, attach incl. remove | X | - | - | Multiple |
| SQLape® | Demographics | Age | Numeric | Years | The expected rates of potentially avoidable readmissions were estimated using the licensed SQLape® tool. Variable specifications can be found online: https://www.bfs.admin.ch/bfs/de/home/statistiken/gesundheit/erhebungen/ms.html. For more information regarding the SQLape algorithm, please check http://www.sqlape.com/readmissions/. Checks whether the patient was admitted urgently (a treatment within 12 hours is indispensable) | - | - | | Multiple |
| | Comorbidities | SQLape diagnosis groups | Categorical | Yes/No | | | - | | Single |
| | | Complexity | Categorical | Simple/ Complex | | - | - | | Single |
| | Interventions/ orders | SQLape surgical intervention groups | Categorical | Yes/No | | - | - | | Single |
| | Resource utilization | Previous hospitalization during the last six months before the index admission date | Categorical | Yes/No | | - | - | | Multiple |
| | | Planned hospitalization | Categorical | Yes/No | | - | - | | Multiple |
| LACE+ | Demographics | Gender (male) | Categorical | Yes/No | Known male gender at the day of hospital admission | - | - | | Multiple |
| | | Age | Numeric | Years | Age at the day of hospital admission | - | - | | Multiple |

(*Continued*)

**Table 1.** (Continued)

| Model | Data category | Variable | Variable type | Unit / categories | Working definition | Adaptations | Differences Cohort A | Differences Cohort B | Data points per observation |
|---|---|---|---|---|---|---|---|---|---|
| | Administrative data | Urgent admission | Categorical | Yes/No | Checks whether the patient was admitted urgently (a treatment within 12 hours is indispensable) | - | - | | Single |
| | | Discharge institution (teaching vs. small non-teaching hospital) | Categorical | Yes/No | Small nonteaching hospital = nonteaching hospital with < 100 beds, large nonteaching hospital = nonteaching hospital ≥ 100 beds | - | - | | Single |
| | | Discharge institution (large vs. small non-teaching hospital) | Categorical | Yes/No | | - | - | | Single |
| | | Number of days on ALC status | Numeric | Days | Alternative level of care (ALC) status stands for patients who stay at the hospital but no longer receive active medical care, coded as main diagnosis Z75.8 ICD-10 code | - | - | | Single |
| | | Current length of stay | Numeric | Days | Counts the number of days of hospitalization of the ongoing stay, starting at the day of inpatient admission. It does consider any time spent in the Emergency Department. Rounded to 3 decimal points. (Time stamp at model calculation—admission time stamp). | - | - | | Multiple |
| | Resource utilization | Number of ED visits in the previous 6 months | Numeric | Visits | A count of the number of ED visits in the last six months. Counts both, those where the patient went home healthy, and the ones, where patients were subsequently submitted to the wards. Lookback starts at the day of admission. | - | - | | Multiple |
| | | Number of urgent admissions in the previous 12 months | Numeric | Visits | A count of the number of urgent hospital admissions (through the ED). The look-back period starts at the day of admission. | - | - | | Multiple |
| | | Number of elective admissions in the previous 12 months | Numeric | Visits | A count of the number of elective hospital admissions, the look-back starts at the day of admission. | - | - | | Multiple |
| | Comorbidity | CMG score | Numeric | Unknown | Case Mix Group (CMG) variable is only available in Canada–not relevant | - | - | | Single |

(*Continued*)

**Table 1.** (Continued)

| Model | Data category | Variable | Variable type | Unit / categories | Working definition | Adaptations | Differences Cohort A | Differences Cohort B | Data points per observation |
|-------|---------------|----------|---------------|-------------------|--------------------|-------------|----------|----------|------------|
| | | Charlson Comorbidity Index (CCI) | Numeric | Points | The Charlson Comorbidity Index was calculated based on all known diagnoses at discharge. A list of exact codes used to identify relevant disorders is available on reasonable request from the corresponding author. | X | - | | Single |
| | Other | Interaction term 1 | Numeric | Points | Age x Charlson Comorbidity Index | - | - | | Single |
| | Other | Interaction term 2 | Numeric | Points | Age x Number of urgent admissions in previous year | - | - | | Multiple |
| | Other | Interaction term 3 | Numeric | Points | Charlson Comorbidity Index x Number of urgent admissions in previous year | - | - | | Single |

(A) Abbreviations: ALC–Alternative Level of Care; ATC–Anatomical Therapeutic Chemical Classification System; BUN—Blood Urea Nitrogen; CCI–Charlson Comorbidity Index; CMG–Case Mix Group; ED–Emergency Department; ECG–Electrocardiography; GM–German modification; ICD–International Statistical Classification of Diseases; LOS–Length of Stay; NSAIDs—Non-Steroidal Anti-Inflammatory drugs; PT/INR—Prothrombin Time and International Normalized Ratio; TARMED–Unified Relative Tariff System.

(B) Notes: The "Adaptation" column shows whether a variable had to be adapted to local regulations, practice patterns or classifications etc.; the "Differences" column indicates, whether in either Cohort A or Cohort B a more prominent discrepancy in variable definition, compared to the derivation study, was present.

higher number of predictors and their nature. Analogous to LACE+, all ICD-10-CM codes were mapped onto ICD-10-GM codes (for the Charlson Comorbidity Index and diagnoses). The mapping table is available from the corresponding author upon request. Five medication subclasses contribute to the Risk of Unplanned Readmission score. An experienced pharmacist developed Swiss-specific therapeutic subgroups of the Anatomical Therapeutic Chemical Classification System (ATC) based on the original specifications and with regard to local regulations and classifications. All subgroups and therein included codes are listed in the S1 Appendix. With respect to the biological data, which are captured differently depending on the site's laboratory system, a laboratory analyst matched all required input parameters with the local system's laboratory components (lab components and applied reference ranges are detailed in S2 Appendix). Finally, the input parameter "restraining order" was excluded because restraining measures were not performed in the period under consideration; remaining order variables were matched with their corresponding "Tarif Medical" (TARMED) service codes (catalog version 1.09, valid from 01.01.2018) as a means of data collection through service recording and billing information [31]. The SQLape® tool was used without any adaptations to produce predictions. Variable specifications can be found online [32, 33]. All adaptations and modifications, as well as differences in variable definition between the original derivation study and this external validation study, are described in detail in S3 Appendix (detailed description of prediction model variables).

This study is based on routinely collected patient data. Therefore, the number of data points per patient available for extraction was dependent on the forgone hospitalization characteristics (i.e., main diagnosis, disease severity, course of disease, and length of stay) and restricted by documentation standards, and/or their compliance, the structural quality of the electronic health record systems (EHRS), and their management [34]. For each predictor, either several

input data points (documented throughout the hospital stay) or a single input data point (documented at admission or discharge) existed (Table 1). In general, the last data point was carried forward [35]. Missing input data required to compute the prediction models were interpreted as follows: missing biological input data (hemoglobin, sodium, etc.) were coded as normal values. Missing comorbidity and medication input data were considered absence of the condition and no active medication, respectively. Missing order input data (imaging or electrocardiogram) were considered as an intervention not ordered. Last, missing records of utilization of healthcare resources (e.g., ED visits, scheduled future admissions, etc.) were considered as nonutilization. This way of dealing with missing values was justified with the common documentation method "Charting by Exception" and is in line with the approach used during the model development. Missing data required to describe patient characteristics (demographics and administrative data) led to patient exclusion from the analysis.

## Sample size

Sample size calculations resulted in an aspired sample size of at least 1000 participants (500 cases and 500 controls) for each site. The proposed sample size is based upon precision. Five hundred cases and 500 controls will ensure that the half-width of a 95% confidence interval for sensitivity and specificity (using frequencies of predicted vs. actual outcome) does not exceed 5%; even for a point estimate of 50%, leading to the widest possible confidence interval, the half-width is supposed to remain slightly below 4.5%. This can be considered an appropriate target precision for the purpose of this study.

## Statistical analysis methods

Model Validation and Comparison: Descriptive analysis was performed for all variables. Categorical variables were described as frequencies (percentages), and continuous variables were described as the means (standard deviations [SDs]) or medians (interquartile ranges [IQRs]), as appropriate. Baseline characteristics (index hospitalization) were compared between unplanned 30-day readmitted and nonreadmitted patients. Differences between groups were tested using binomial, Pearson's $\chi 2$, or Wilcoxon rank-sum tests, as appropriate. The unit of analysis was hospitalization.

For Cohort A, the LACE+ and the Epic Unplanned Readmission model predictions were calculated as the inverse of $1 + e^{-(intercept + \beta_1 * x_1 + \cdots + \beta_K * x_K)}$, where $\beta$ is the regression coefficient of each covariate (x) and K is the total number of covariates. SQLape® readmission probabilities were calculated by the SQLape® tool [33]. All information on model variables was either collected from published derivation studies or provided by the developers directly.

To assess performance, the traditional statistical approach is to quantify how close predictions are to the actual outcome. As an overall performance measure, composed of discrimination and calibration, the Brier score was calculated [36]. Assessed separately, the discriminative ability was measured using Harrel's C-statistic, which, for binary outcomes, is identical to the area under the receiver operating characteristic (ROC) curve (AUC). For calibration, the Hosmer-Lemeshow goodness-of-fit test was graphically illustrated by plotting the predicted outcomes by decile against the observations [29]. Furthermore, another novel performance measure, category-based net reclassification improvement (NRI) was computed [37, 38]. The NRI separately considers individuals who develop and who do not develop the event of interest and therefore provides additional information not available from the AUC. The NRI is defined as the sum of the net proportions of correctly reclassified patients with and without the outcome [37]. To be able to compare with the results of the derivation studies, performance analysis was performed based on the latest input data points available, but no later than 8 a.m. on the day of discharge.

Cohort B analysis: The descriptive analysis was performed analogous to the analysis ran for Cohort A. Supplementary, to test the comparability of Cohort A and Cohort B, a set of patient characteristics was compared using binomial, Pearson's $\chi 2$, or Wilcoxon rank-sum tests, as appropriate.

For Cohort B, the Epic Unplanned Readmission model predictions were calculated by the Epic Electronic Health Record "AI & Analytics" module. The performance assessment was conducted analogous to the evaluation performed for Cohort A but it focused only on the Epic prediction model. To provide better insight into the predictive ability at different times throughout the hospital stay, subgroup analyses were performed. The Epic prediction model's performance was assessed for the admission day, first to fifth day of hospitalization, day before discharge, and discharge day. For each day, the predictive ability was assessed for input data points available at 8 a.m. and 12 a.m. and presented in a forest plot. All statistical analyses were performed in RStudio, Version 1.2.5019 and STATA/SE, Version 16.0.

## Risk groups

Based on the recommendation of the developers of the Epic model, patients were divided into four risk groups using the organization's current unplanned readmission rate as baseline risk: "no risk = 0 –baseline", "low risk = baseline– 2 x baseline", "medium risk = 2 x baseline– 3 x baseline", and "high risk = > 3 x baseline".

## Development vs. validation

The characteristics of the derivation cohorts are summarized in Table 2 (Model transportability–summary characteristics). The readmission rate of this study's cohort was approximately 4.7%, whereas it ranged between 5.2 and 16.9% in the derivation cohorts. With regard to the outcome of all derivation studies, a distinction was made between planned and unplanned readmissions. Knowing that some treatments require repeated hospitalization (e.g., multi-course treatments such as chemotherapy), readmissions by definition were no true indicator of quality. On that account, the LACE+ study excluded readmission that was foreseeable, i.e., nonurgent. The SQLape® and Epic model operationalized the exclusion of planned readmissions by applying a more complex decision rule. This rule checks for specific procedure and diagnosis categories that are usually considered planned, acute, or complication of care [8, 24]. A distinction was also made between avoidable and nonavoidable readmissions. This validation study followed the approach taken by the LACE+ derivation study. All predictors were defined in line with the derivation studies, except for the following: the "Current length of hospital stay" considered the time spent in the ED, while the derivation study did not; the "Numbers of active medication orders" considered all active medications throughout the hospitalization, while the derivation study considered only the active medications at the day of score calculation; the "Diagnoses and comorbidities" were based on the discharge diagnosis instead of the known diagnosis at the day of score calculation; see Table 1 (Model predictors), column "Differences", and S3 Appendix (Detailed description of prediction model variables). All beforementioned differences hold true only for the baseline cohort. Cohort B was not affected by actual differences but by setting specific adaptations.

## Results

### Participants

During the study period, a total of 53,497 discharges were recorded. All discharges of 2018 were grouped as Cohort A, including 42,381 records; discharges of the last quarter in 2019

**Table 2. Model transportability–summary characteristics.**

| Model | Setting | Prevalence* | Exclusion criteria | Outcome | Development data | Missing data treatment | Point in time of score calculation | Differences in predictor variables** | Cohort A | Cohort B |
|---|---|---|---|---|---|---|---|---|---|---|
| Epic Risk of Unplanned Readmission model | U.S. 26 acute care academic, regional, and community hospitals; 275,000 medical and surgical encounter | 16.9% | Patients younger than 1 year, and older than 100 years; patients resided outside of the United States; deceased patients; patients who were transferred directly to another hospital; patients being hospitalized for primary psychiatric diseases, or medical treatment of cancer. | Unplanned readmission within 30 days | Hospital data from the year 2016; did not consider external readmissions | Carried forward last value; Interpreted e.g. missing biological value as normal lab results, missing order values as no intervention performed, etc. | (Highest score at) discharge day | Demographics | - | - |
| | | | | | | | | Administrative data | X | - |
| | | | | | | | | Resource utilization | - | - |
| | | | | | | | | Medications | X | - |
| | | | | | | | | Comorbidities | X | - |
| | | | | | | | | Biological data | - | - |
| | | | | | | | | Interventions/ orders | - | - |
| SQLape® | CH 49 Swiss hospitals (including academic and general hospitals); 131,809 medical and surgical encounter | 5.2%* | Healthy newborns; residents outside of Switzerland; elective surgical stays that were performed as day surgery****; Psychiatric, geriatric, palliative, and rehabilitative patients; patients being directly transferred to another hospital after admission; deceased patients; patients with sleep apnea | Potentially avoidable readmissions within 30 days | Hospital data from the year 2000; considered external readmissions*** | Complete-case analysis | Discharge day | Demographics | - | N/A |
| | | | | | | | | Comorbidities | - | N/A |
| | | | | | | | | Resource utilization | - | N/A |
| | | | | | | | | Interventions/ orders | - | N/A |

(*Continued*)

**Table 2.** (Continued)

| Model | Setting | Prevalence* | Exclusion criteria | Outcome | Development data | Missing data treatment | Point in time of score calculation | Differences in predictor variables** | Cohort A | Cohort B |
|---|---|---|---|---|---|---|---|---|---|---|
| LACE+ | CA Acute care Ontario hospitals; 500'000 medical and surgical patients | 6.1% | Discharges to rehabilitation and long-term care facilities; same-day surgeries, psychiatric and obstetric patients; patients who were ineligible for health care coverage in Ontario | Unplanned, i.e. urgent readmission within 30 days | Insurance data from the year 2003–2009; considered external readmissions*** | No missing values | Discharge day | Demographics | - | N/A |
| | | | | | | | | Administrative data | - | N/A |
| | | | | | | | | Resource utilization | - | N/A |
| | | | | | | | | Comorbidities | - | N/A |
| External validation | CH General hospital; medical and surgical inpatients; Cohort A: 28,304. Cohort B: 7080 | Cohort A: 5.1% Cohort B: 4.3% | Admissions/transfers from another psychiatric, rehabilitative or acute care ward from the same hospital; patients discharged to a destination other than the patient's home; patients with a foreign or unknown residence; deceased before discharge | Unplanned readmission within 30 days, An unplanned readmission was defined as an readmission not scheduled in advance that requires treatment within 12 hours | Hospital data from the year 2018–2020; did not consider external readmissions | Carried forward last value; Interpreted e.g. missing biological value as normal lab results, missing order values as no intervention performed, etc. | Cohort A: score at the discharge day. Cohort B: 8 and 12 a.m. at admission day, 1st to 5th day, day before discharge and discharge day | Please see above and Table 1. (Model predictors) | | |

(A) Abbreviations: CA–Canada; CH–Switzerland; U.S.–United States.

(B) * Prevalence of the event of interest (unplanned readmissions within 30 days); for the SQLape® the prevalence represents the rate of potentially avoidable readmissions; recent studies reported a broad range for the proportion of unplanned 30-day readmissions deemed potentially avoidable (23.1%, 95% CI, 21.7% - 24.5%) [39–41].

(C) ** Please see Table 1. (Model predictors) for a listing of all individual predictor variables.

(D) *** External readmission were considered to some extent, within given boundaries (e.g. provinces, cantons, clinic networks etc.).

(E) **** Identification of patients that qualify for one-day surgery [42, 43].

(F) N/A, the LACE+ and the SQLape® were not computed for Cohort B.

were grouped as Cohort B, including 11,116 records. After exclusions, Cohorts A and B comprised 28,112 and 7071 records, respectively (see Fig 1. Flow Chart). For 23,116 records (82.2%) of Cohort A, data sufficed to replicate the LACE+, SQLape®, and Epic Risk of Unplanned Readmission scores for the day of discharge at 8 a.m. (for 28,112 records, LACE + and Epic scores were available). For each of the 7071 records in Cohort B, numerous Epic risk scores were exported from the EHR "AI & Analytics" module. The number of scores was dependent on the LOS, admission, and discharge time. In total, 30,187 discharges with scores were included in the analysis (Cohorts A + B). Of Cohort A, out of 23,116 inpatients, 1181

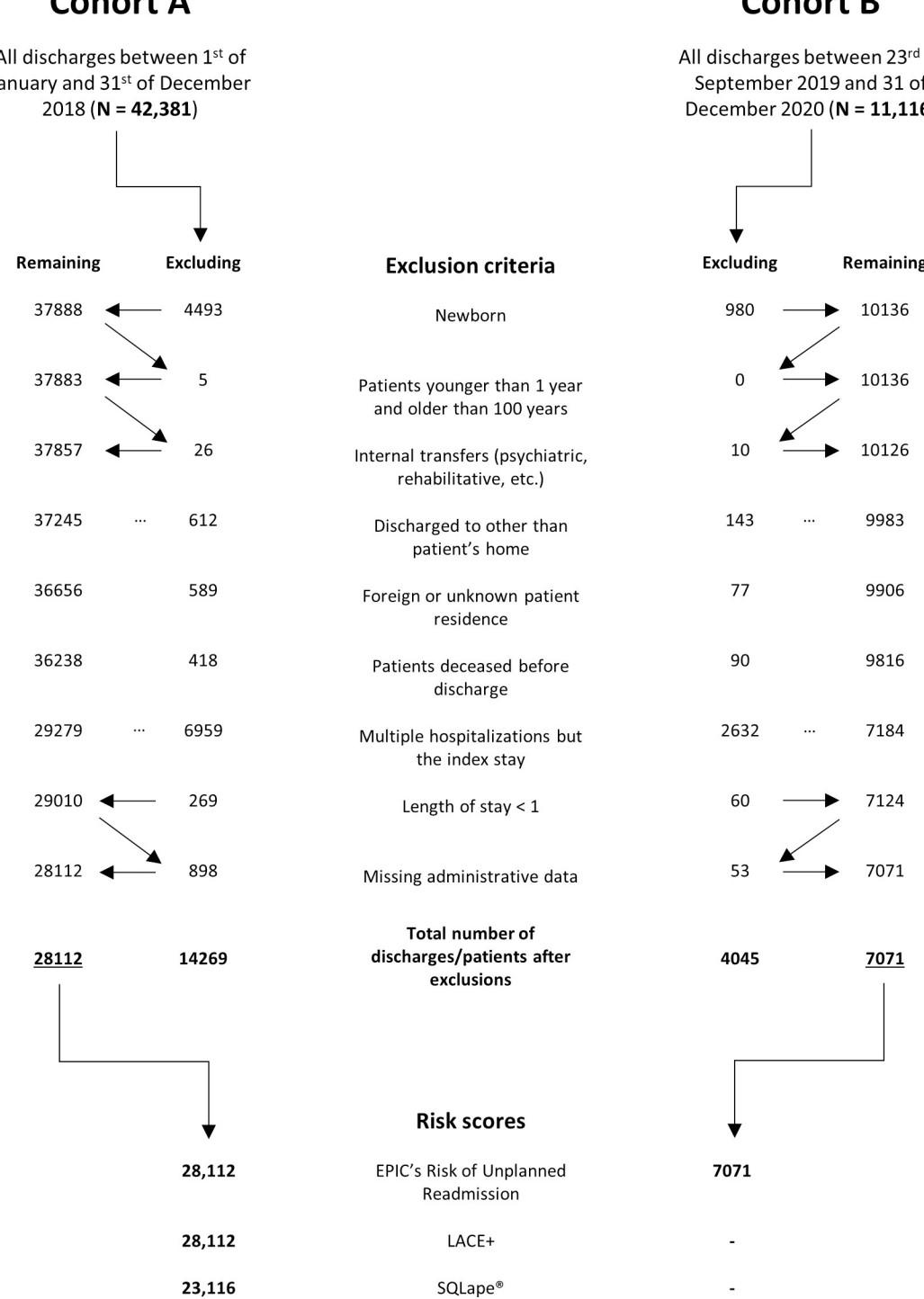

## Cohort A

All discharges between 1st of January and 31st of December 2018 (**N = 42,381**)

## Cohort B

All discharges between 23rd of September 2019 and 31 of December 2020 (**N = 11,116**)

| Remaining | Excluding | Exclusion criteria | Excluding | Remaining |
|---|---|---|---|---|
| 37888 | 4493 | Newborn | 980 | 10136 |
| 37883 | 5 | Patients younger than 1 year and older than 100 years | 0 | 10136 |
| 37857 | 26 | Internal transfers (psychiatric, rehabilitative, etc.) | 10 | 10126 |
| 37245 | ⋯ 612 | Discharged to other than patient's home | 143 ⋯ | 9983 |
| 36656 | 589 | Foreign or unknown patient residence | 77 | 9906 |
| 36238 | 418 | Patients deceased before discharge | 90 | 9816 |
| 29279 | ⋯ 6959 | Multiple hospitalizations but the index stay | 2632 ⋯ | 7184 |
| 29010 | 269 | Length of stay < 1 | 60 | 7124 |
| 28112 | 898 | Missing administrative data | 53 | 7071 |
| **28112** | 14269 | **Total number of discharges/patients after exclusions** | 4045 | **7071** |

### Risk scores

| | | | | |
|---|---|---|---|---|
| **28,112** | | EPIC's Risk of Unplanned Readmission | | **7071** |
| **28,112** | | LACE+ | | - |
| **23,116** | | SQLape® | | - |

**Fig 1. Flow chart.** (A) After the exclusion of all hospitalizations but the index hospitalization, discharge numbers equal the number of distinct inpatients.

(5.1%) were readmitted within 30 days of the index discharge date. Of Cohort B, 303 inpatients were readmitted. This corresponds to 4.3% of the 7071 inpatients (see Table 3. Score schedule).

**Table 3. Score schedule.**

| | Cohort A | | | Cohort B | | |
|---|---|---|---|---|---|---|
| | Jan. 01, 2018 –Dec. 31, 2018 | | | Oct. 01, 2019 –Dec. 31, 2019 | | |
| Scores (day, time) | Patients/ scores | without readmission (%) | with readmission (%) | Patients/ scores | without readmission (%) | with readmission (%) |
| **LACE+ scores**, discharge day 8 a.m. | 28,112 | 26,797 (95.3) | 1315 (4.7) | - | - | - |
| **SQLape® scores**, discharge day 8 a.m. | 23,116 | 21,935 (94.9) | 1181 (5.1) | - | - | - |
| **Epic score**, admission day (8 a.m.) | - | - | - | 1233 | 1201 (97.4) | 32 (2.6) |
| **Epic score**, admission day (12 a.m.) | - | - | - | 3217 | 3120 (97.0) | 97 (3.0) |
| **Epic score**, 1st day (8 a.m.) | - | - | - | 6787 | 6510 (95.9) | 277 (4.1) |
| **Epic score**, 1st day (12 a.m.) | - | - | - | 6567 | 6289 (95.8) | 278 (4.2) |
| **Epic score**, 2nd day (8.a.m.) | - | - | - | 5935 | 5676 (95.6) | 259 (4.4) |
| **Epic score**, 2nd day (12 a.m.) | - | - | - | 5233 | 5000 (95.6) | 233 (4.4) |
| **Epic score**, 3rd day (8 a.m.) | - | - | - | 4273 | 4074 (95.3) | 199 (4.7) |
| **Epic score**, 3rd day (12 a.m.) | - | - | - | 3707 | 3523 (95.0) | 184 (5.0) |
| **Epic score**, 4th day (8 a.m.) | - | - | - | 2976 | 2814 (94.6) | 162 (5.4) |
| **Epic score**, 4th day (12 a.m.) | - | - | - | 2593 | 2441 (94.1) | 152 (5.9) |
| **Epic score**, 5th day (8 a.m.) | - | - | - | 2100 | 1966 (93.6) | 134 (6.4) |
| **Epic score**, 5th day (12 a.m.) | - | - | - | 1875 | 1751 (93.4) | 124 (6.6) |
| **Epic score**, day before discharge (8 a.m.) | - | - | - | 6259 | 5986 (95.6) | 273 (4.4) |
| **Epic score**, day before discharge (12 a.m.) | - | - | - | 6530 | 6250 (95.7) | 280 (4.3) |
| **Epic score**, discharge day (8 a.m.) | 28,112 | 26,797 (95.3) | 1315 (4.7) | 7071 | 6768 (95.7) | 303 (4.3) |
| **Epic score**, discharge day (12 a.m.) | - | - | - | 4234 | 4047 (95.6) | 187 (4.4) |

The baseline characteristics of both cohorts are summarized in Table 4 (Baseline characteristics) as per the occurrence of the event of interest and according eligibility. After exclusions, the mean age (SD) was 51 years (24) in both cohorts; between 53 and 54% were female; and the average LOS (SD) was 4.8 (5.3) and 4.6 (5.3) days, respectively. Compared to Cohort A, Cohort B consisted of a higher proportion of surgical and fewer medical inpatients. In contrast, average LOS was shorter. Overall, inpatients with an unplanned readmission were older, had an urgent index admission more often, had a longer LOS, were more severely sick (according to the CMI), and had higher risk scores. All these differences were statistically significant (P < 0.05). Finally, Table 5 shows the characteristics of the predictor variables of Cohort A, grouped by patients with and without an unplanned readmission.

## Overall performance

To quantify the overall performance, the Brier score was used. It is a quadratic scoring rule, where the squared difference between the actual binary outcome and the predictions are calculated. The Brier score can range from 0 for a perfect model to 0.25 for a noninformative model (the lower the better) [37]. The Brier scores were as follows: Epic model– 0.0484, LACE+– 0.0474, and SQLape® – 0.0473 (based on the maximum number of available scores of Cohort A, N = 28,112, Brier scores were: Epic model—0.0457 and LACE+ - 0.0437, respectively). The Brier score for the Epic model on the day of discharge at 8 a.m., based on Cohort B, was 0.0414 (other Brier scores are detailed in S4 Appendix, Cohort B–Brier scores). According to the student's t-test results, only the Epic model yielded a significantly different Brier score (p<0.001) compared to LACE+ and SQLape®.

**Table 4. Baseline characteristics.**

| Variable | Cohort A | | | | | Cohort B | | | | |
|---|---|---|---|---|---|---|---|---|---|---|
| | Jan. 01, 2018 –Dec. 31, 2018 | | | | | Oct. 01, 2019 –Dec. 31, 2019 | | | | |
| | Total before exclusion (N = 42,381) | Total after exclusion with scores (N = 23,116) | With readmission (N = 1181)* | Without readmission (N = 21,935) | p-value | Total before exclusion (N = 11,204) | Total after exclusion with score (N = 7071) | With readmission (N = 303)** | Without readmission (N = 6768) | p-value |
| **Age**, years, mean (±SD) | 48 (±28.3) | 51 (±24.2) | 60 (±23.1) | 51 (±24.1) | < 0.001 | 48 (±28.1) | 51 (±24.0) | 58 (±23.4) | 51 (±24.0) | < 0.001 |
| **Female**, n (%) | | | | | < 0.001 | | | | | 0.0056 |
| Male | 20,748 (49.0%) | 10,605 (45.9%) | 639 (54.1%) | 9966 (45.4%) | | 5570 (49.7%) | 3304 (46.7%) | 165 (54.5%) | 3139 (46.4%) | |
| Female | 21,636 (51.0%) | 12,511 (54.1%) | 542 (45.9%) | 11,9669 (54.6%) | | 4366 (50.3%) | 3767 (53.3%) | 138 (45.5%) | 3629 (53.6%) | |
| **Insurance**, n (%) | | | | | 0.7243 | | | | | 0.2654 |
| General | 35,057 (82.7%) | 18,852 (81.6%) | 959 (81.2%) | 17,893 (81.6%) | | 9142 (81.6%) | 5638 (79.7%) | 249 (82.2%) | 5389 (79.6%) | |
| Semi-private/ Private | 7324 (17.3%) | 4264 (18.4%) | 222 (18.8%) | 4042 (18.4%) | | 2062 (18.4%) | 1433 (20.3%) | 54 (17.8%) | 1379 (20.4%) | |
| **Origin of patient**, n (%) | | | | | < 0.001 | | | | | 0.2445 |
| Home | 37,998 (89.7%) | 21,457 (92.8%) | 1077 (91.2%) | 20,380 (92.9%) | | 10,359 (92.5%) | 6591 (93.2%) | 282 (93.1%) | 6309 (93.2%) | |
| Nursing home | 1120 (2.6%) | 564 (2.4%) | 58 (4.9%) | 506 (2.3%) | | 320 (2.9%) | 154 (2.2%) | 10 (3.3%) | 144 (2.1%) | |
| Other | 3263 (7.7%) | 1095 (4.8%) | 46 (3.9%) | 1227 (4.8%) | | 525 (4.6%) | 326 (4.6%) | 11 (3.6%) | 315 (4.7%) | |
| **Admission type**, n (%) | | | | | < 0.001 | | | | | < 0.001 |
| Urgent | 20,882 (49.3%) | 12,279 (53.1%) | 804 (68.1%) | 11,475 (52.3%) | | 5569 (49.7%) | 3487 (49.3%) | 196 (64.7%) | 3291 (48.6%) | |
| Elective | 17,284 (40.8%) | 10,652 (46.1%) | 369 (31.2%) | 10,283 (46.9%) | | 4657 (41.6%) | 3494 (49.4%) | 101 (33.3%) | 3393 (50.1%) | |
| Other | 4215 (9.9%) | 185 (0.8%) | 8 (0.7%) | 177 (0.8%) | | 978 (8.7%) | 90 (1.3%) | 6 (2.0%) | 84 (1.3%) | |
| **Discharge destination**, n (%) | | | | | < 0.001 | | | | | 0.0012 |
| Patient's home | 35,321 (83.3%) | 21,286 (92.1%) | 987 (83.6%) | 20,299 (92.5%) | | 9519 (85.0%) | 6340 (89.7%) | 259 (85.5%) | 6081 (89.8%) | |
| Nursing home | 2497 (5.9%) | 1399 (6.0%) | 138 (11.7%) | 1261 (5.8%) | | 634 (5.7%) | 368 (5.2%) | 29 (9.6%) | 339 (5.0%) | |
| Other | 4563 (10.8%) | 431 (1.9%) | 56 (4.7%) | 375 (1.7%) | | 1051 (9.3) | 363 (5.1%) | 15 (4.9%) | 348 (5.2%) | |
| **ALOS**, day (± SD) | 6.0 (±7.9) | 4.8 (±5.3) | 7.3 (±8.2) | 4.6 (±5.0) | < 0.001 | 5.1 (±6.5) | 4.6 (±5.3) | 6.8 (±7.4) | 4.5 (±5.2) | < 0.001 |
| **CMI**, mean (± SD) | 1.123 (±1.540) | 1.057 (±1.032) | 1.403 (±1.585) | 1.038 (±0.991) | < 0.001 | 1.093 (±1.376) | 1.070 (±1.135) | 1.261 (±1.152) | 1.061 (±1.133) | 0.0050 |
| **Specialty**, n (%) | | | | | < 0.001 | | | | | < 0.001 |
| General Internal Medicine | 11,893 (28.1%) | 6506 (28.1%) | 513 (43.4%) | 5993 (27.3%) | | 2133 (19.0%) | 1260 (17.8%) | 77 (25.5%) | 1183 (17.5%) | |
| General Surgery | 14,607 (34.5%) | 9245 (40.0%) | 445 (37.7%) | 8800 (40.1%) | | 5032 (44.9%) | 3487 (49.3%) | 152 (50.2%) | 3335 (49.3%) | |
| Gynecology | 7818 (18.4%) | 3771 (16.3%) | 102 (8.6%) | 3669 (16.7%) | | 2082 (18.6%) | 1079 (15.3%) | 31 (10.2%) | 1048 (15.5%) | |
| Pediatric | 4272 (10.1%) | 1885 (8.2%) | 54 (4.6%) | 1831 (8.3%) | | 1091 (9.7%) | 612 (8.7%) | 15 (4.9%) | 597 (8.8%) | |
| Ophthalmology | 1477 (3.5%) | 848 (3.7%) | 26 (2.2%) | 822 (3.8%) | | 432 (3.9%) | 350 (4.9%) | 10 (3.3%) | 340 (5.0%) | |
| Oto-Rhino-Laryngology | 1465 (3.5%) | 861 (3.7%) | 41 (3.5%) | 820 (3.8%) | | 342 (3.0%) | 283 (4.0%) | 18 (5.9%) | 265 (3.9%) | |

*(Continued)*

**Table 4.** (Continued)

| Variable | Cohort A | | | | | Cohort B | | | | |
|---|---|---|---|---|---|---|---|---|---|---|
| | Jan. 01, 2018 –Dec. 31, 2018 | | | | | Oct. 01, 2019 –Dec. 31, 2019 | | | | |
| | Total before exclusion (N = 42,381) | Total after exclusion with scores (N = 23,116) | With readmission (N = 1181)* | Without readmission (N = 21,935) | p-value | Total before exclusion (N = 11,204) | Total after exclusion with score (N = 7071) | With readmission (N = 303)** | Without readmission (N = 6768) | p-value |
| Other | 849 (1.9%) | - | - | - | | 92 (0.9%) | - | - | - | |
| **Epic Risk of Unplanned Readmission** | - | 0.0834 (±0.0524) | 0.1210 (±0.0787) | 0.0814 (±0.0498) | < 0.001 | - | 0.0725 (±0.0380) | 0.0939 (±0.0446) | 0.0715 (±0.0374) | < 0.001 |
| **SQLape®** | - | 0.0307 (±0.0284) | 0.0536 (±0.0368) | 0.0295 (±0.0274) | < 0.001 | - | - | - | - | |
| **LACE+** | - | 0.0294 (±0.0329) | 0.0547 (±0.0504) | 0.0280 (±0.0311) | < 0.001 | - | - | - | - | |

(A) Abbreviations: ALOS–Average Length of Stay; CMI–Case Mix Index; SD–Standard Deviation.

(B) * Cohort A: Prevalence of the event of interest (unplanned readmissions within 30 days) = 5.1%.

(C) ** Cohort B: Prevalence of the event of interest (unplanned readmissions within 30 days) = 4.3%.

(D) Scores (Epic score, SQLape® and LACE+) are reported as mean values (SD).

(E) P values are defined as the probability under the assumption of no difference (null hypothesis), of obtaining a proportion different from what was observed in subjects without a readmission.

## Calibration

For calibration, the Hosmer-Lemeshow goodness-of-fit test was graphically illustrated by plotting the predicted risk by deciles (and risk group thresholds) against the observations. The diagonal line is the line of perfect calibration, described with an intercept alpha of 0 and slope of 1. The graph indicates that the Epic model had a poor fit and generally overestimated the observed probability, especially at higher deciles of risk (Table 6). The intercept, which relates to the calibration-in-the-large (CITL), was -0.542, and the slope was 1.105. The SQLape® and LACE+ showed very similar results but underestimated at higher deciles of risk. The SQLape® intercept was 0.550, and the slope was 0.759; the intercept and slope of LACE+ were 0.605 and 0.798, respectively. The p-values of the Hosmer-Lemeshow $\chi^2$ statistic were p<0.001 for all three models. Calibration plots by decile are presented in Fig 2 (calibration plots Cohort A), and calibration plots by risk group thresholds are accessible as S5 Appendix. Calibration plots based on Cohort B were computed and are illustrated in S6 Appendix.

## Discrimination

In theory, the AUC ranges between 0.5 and 1.0. The AUCs for the risk scores based on Cohort A on the day of discharge at 8 a.m. were as follows: Epic AUC 0.692 (95% CI 0.676–0.708), LACE+ index AUC 0.703 (95% CI 0.687–0.719), and SQLape® AUC 0.705 (95% CI 0.690–0.720). Neither the LACE+ nor the Epic model yielded a significantly different AUC than that of the SQLape® (p>0.05). The ROC curves are presented in Fig 3. Using the maximum number of available scores of Cohort A (N = 28,112) did not lead to a significant change in AUC (Epic model: 0.680, 95% CI 0.664–0.696; LACE+: 0.693, 95% CI 0.677–0.709; p<0.05).

The predictive ability of the Epic Risk of Unplanned Readmission model was also assessed at different times throughout the hospital stay based on all records of Cohort B. The AUCs ranged between 0.527 and 0.677. Using the AUC on the day of discharge at 8 a.m. as a

**Table 5. Baseline characteristics—predictor variables.**

| Variables | Cohort A | | | |
| --- | --- | --- | --- | --- |
| | Jan. 01, 2018 –Dec. 31, 2018 | | | |
| | Total after exclusion (N = 23,116) | Readmission (N = 1181) | No Readmission (N = 21,935) | p-value |
| **Age**, years, mean (±SD) | 51 (±24.2) | 60 (±23.1) | 51 (±24.1) | < 0.001 |
| **Gender–male**, n (%) | | | | |
| yes | 10,605 (45.9%) | 639 (54.0%) | 9966 (45.4%) | < 0.001 |
| **Current length of stay**, mean (±SD) | 4.7 (±5.2) | 7.2 (±8.2) | 4.6 (±5.0) | < 0.001 |
| **Urgent admission**, n (%) | | | | |
| yes | 12,279 (53.1%) | 1181 (100%) | 11,475 (52.3%) | < 0.001 |
| **Number of past ED visits, in the last 6 months**, n (%) | | | | < 0.001 |
| 0 | 20,478 (88.6%) | 1001 (84.7%) | 19,477 (88.8%) | |
| 1 | 2062 (8.9%) | 125 (10.6%) | 1937 (8.8%) | |
| 2 | 429 (1.9%) | 35 (3.0%) | 394 (1.8%) | |
| 3 | 97 (0.4%) | 13 (1.1%) | 84 (0.4%) | |
| >3 | 50 (0.2%) | 7 (0.6%) | 43 (0.2%) | |
| **Number of past admissions, in the last 12 months**, n (%) | | | | < 0.001 |
| 0 | 20,295 (87.8%) | 947 (80.2%) | 19,348 (88.2%) | |
| 1 | 2023 (8.8%) | 133 (11.3%) | 1890 (8.6%) | |
| 2 | 500 (2.2%) | 50 (4.2%) | 450 (2.1%) | |
| 3 | 175 (0.8%) | 24 (2.1%) | 151 (0.7%) | |
| >3 | 123 (0.5%) | 27 (2.2%) | 96 (0.4%) | |
| **Number of urgent admissions, in the last 12 months**, n (%) | | | | < 0.001 |
| 0 | 22,714 (98.3%) | 1129 (95.6%) | 21,585 (98.4%) | |
| 1 | 331 (1.4%) | 39 (3.3%) | 292 (1.3%) | |
| >1 | 71 (0.3%) | 13 (1.1%) | 58 (0.3%) | |
| **Number of elective admissions, in the last 12 months**, n (%) | | | | - |
| 0 | 23,107 (99.9%) | 1181 (100.0%) | 21,926 (100.0%) | |
| >0 | 9 (0.1%) | 0 | 9 (0.1%) | |
| **Number of days on ALC status**, n (%) | | | | |
| 0 | 23,116 (100.0%) | 1181 (100.0%) | 21,935 (100.0%) | |
| **Has future scheduled appointments**, n (%) | | | | |
| yes | 743 (3.2%) | 24 (2.0%) | 719 (3.3%) | 0.0114 |
| **Prior length of 10 days or more in the last 12 months**, n (%) | | | | |
| yes | 729 (3.2%) | 88 (7.5%) | 641 (2.9%) | < 0.001 |
| **Diagnosis of cancer**, n (%) | | | | |
| yes | 2092 (9.1%) | 263 (22.3%) | 1829 (8.3%) | < 0.001 |
| **Diagnosis of Deficiency Anemia**, n (%) | | | | |
| yes | 644 (2.8%) | 88 (7.5%) | 556 (2.5%) | < 0.001 |
| **Diagnosis of Renal Failure**, n (%) | | | | |

*(Continued)*

**Table 5.** (Continued)

| Variables | Cohort A | | | |
|---|---|---|---|---|
| | Jan. 01, 2018 –Dec. 31, 2018 | | | |
| | Total after exclusion (N = 23,116) | Readmission (N = 1181) | No Readmission (N = 21,935) | p-value |
| yes | 1737 (7.5%) | 226 (19.1%) | 1511 (6.9%) | < 0.001 |
| **Diagnosis of Drug Abuse**, n (%) | | | | |
| yes | 191 (0.8%) | 11 (0.9%) | 180 (0.8%) | 0.6296 |
| **Diagnosis of Electrolyte disorder**, n (%) | | | | |
| yes | 1903 (8.2%) | 296 (25.1%) | 1607 (7.3%) | < 0.001 |
| **Hemoglobin–low**, n (%) | | | | |
| yes | 4451 (19.3%) | 413 (35.0%) | 4038 (18.4%) | < 0.001 |
| **Calcium–low**, n (%) | | | | |
| yes | 793 (3.4%) | 65 (5.5%) | 728 (3.3%) | < 0.001 |
| **Blood Urea Nitrogen (BUN)–high**, n (%) | | | | |
| yes | 522 (2.3%) | 51 (4.3%) | 471 (2.1%) | < 0.001 |
| **Creatinine–high**, n (%) | | | | |
| yes | 1856 (8.0%) | 174 (14.7%) | 1682 (7.7%) | < 0.001 |
| **Phosphate–tested**, n (%) | | | | |
| yes | 894 (3.9%) | 68 (5.8%) | 826 (3.8%) | 0.0010 |
| **Prothrombin Time and International Normalized Ratio (INR)—high**, n (%) | | | | |
| yes | 77 (0.3%) | 7 (0.6%) | 70 (0.3%) | 0.0971 |
| **Anticoagulants**, n (%) | | | | |
| yes | 16,148 (69.9%) | 933 (79.0%) | 15,215 (69.4%) | < 0.001 |
| **Non-Steroidal Anti-Inflammatory drugs (NSAIDs)**, n (%) | | | | |
| yes | 9164 (39.6%) | 309 (26.2%) | 8855 (40.4%) | < 0.001 |
| **Corticosteroids**, n (%) | | | | |
| yes | 7465 (32.3%) | 400 (33.9%) | 7065 (32.2%) | 0.2246 |
| **Antipsychotics**, n (%) | | | | |
| yes | 1733 (7.5%) | 163 (13.8%) | 1570 (7.2%) | < 0.001 |
| **Ulcer medication**, n (%) | | | | |
| yes | 8547 (37.0%) | 564 (47.8%) | 7983 (36.4%) | < 0.001 |
| **Number of active medication orders**, mean (±SD) | 22 (±14.3) | 27 (±18.1) | 22 (±14.1) | < 0.001 |
| **Imaging orders**, n (%) | | | | |
| yes | 19,467 (84.2%) | 1054 (89.2%) | 18,413 (83.9%) | < 0.001 |
| **Electrocardiography (ECG)**, n (%) | | | | |
| yes | 8586 (37.1%) | 663 (53.3%) | 7923 (36.1%) | < 0.001 |
| **Charlson Comorbidity Index (CCI)**, EPIC adapted version, mean (±SD) | 1.0 (±2.1) | 2.6 (±3.2) | 0.9 (±2.0) | < 0.001 |

*(Continued)*

**Table 5.** (Continued)

| Variables | Cohort A | | | |
|---|---|---|---|---|
| | Jan. 01, 2018 –Dec. 31, 2018 | | | |
| | Total after exclusion (N = 23,116) | Readmission (N = 1181) | No Readmission (N = 21,935) | p-value |
| **Charlson Comorbidity Index (CCI)**, mean (±SD) | 0.8 (±1.9) | 2.0 (±2.9) | 0.8 (±1.8) | < 0.001 |

(A) Abbreviations: ALC–Alternative level of care; ED–Emergency Department; SD–Standard Deviation.

(B) P values are defined as the probability under the assumption of no difference (null hypothesis), of obtaining a proportion different from what was observed in subjects without a readmission.

reference, the Epic model yielded significantly different AUCs only compared to the scores computed on the admission day (p<0.024). All AUCs are presented in Fig 4.

## Reclassification

Using SQLape® as a quasi standard for predicting the risk of unplanned readmission in Swiss inpatient populations, the category-based net reclassification improvement (NRI) for the LACE+ and the Epic Risk of Unplanned Readmission model was computed. For LACE+, the NRI for events was 1.01%, and the NRI for nonevents was 3.27%; for the Epic model, the NRI for events was 71.54%, and the NRI for nonevents was -67.24%. The sum of both components resulted in an overall NRI of 0.042 for LACE+ and 0.043 for the Epic model. The category-based NRI components can be interpreted as net percentages of persons with or without events correctly reclassified. Negative percentages are interpreted as a net worsening in risk classification (NRI components range between -100% and +100%). However, the overall category-based NRI is a statistic that is implicitly weighted for the event rate and cannot be interpreted as a percentage. Its theoretical range is -2 to +2 [38]. Reclassification tables are accessible as S7 Appendix.

## Discussion

### Limitations

This study has several important limitations that need to be addressed. All limitations primarily but not exclusively relate to Cohort A. First, this study is a single-center study shaped by patient characteristics, local practice patterns, and EHR systems in place. Therefore, the findings may not be generalizable to all Swiss hospitals, particularly concerning university hospitals, whose patient populations often differ in characteristics, and organizations outside of Switzerland, where certain data points might be captured differently, or not at all. Even when specific data points exist, their distribution might vary (e.g., emergency department utilization, medications, etc.). Second, although this study was primarily about the validation of the Epic model, its comparison with the quasi standard SQLape® required the application of exclusion criteria different from criteria used in each individual derivation study but appropriate for the Swiss setting (see Table 2 Model transportability–summary characteristics). Discrepancies were minor in regard to the LACE+ model but more significant concerning the Epic model. This could have introduced selection bias, resulting in lower model accuracy (compared to the derivation study), with model predictions over- or underestimating the actual risk. Third, only readmissions to the same hospital were considered. According to a survey of the Swiss National Association for Quality Development in Hospitals and Clinics, external readmissions

**Table 6. Observed vs. predicted 30-day unplanned readmissions.**

| Model | Risk | Risk category | Patients (%) | Observed proportion (%) | Predicted proportion (%) |
|---|---|---|---|---|---|
| **Epic model** | [0–0.051] | No risk | 5009 (22) | 2.2 | 4.4 |
| | (0.051–0.102] | Low risk | 12,998 (56) | 3.8 | 7.1 |
| | (0.102–0.153] | Medium risk | 3443 (15) | 8.6 | 12.2 |
| | (0.153–1] | High risk | 1666 (7) | 16.7 | 22.2 |
| | [0.02837–0.04398] | Decile 1 | 2312 (10) | 1.9 | 3.9 |
| | (0.04398–0.05018] | 2 | 2312 (10) | 2.4 | 4.7 |
| | (0.05018–0.05543] | 3 | 2311 (10) | 2.6 | 5.3 |
| | (0.05544–0.06138] | 4 | 2312 (10) | 3.0 | 5.8 |
| | (0.06138–0.06860] | 5 | 2311 (10) | 3.5 | 6.5 |
| | (0.06860–0.07680] | 6 | 2312 (10) | 3.0 | 7.3 |
| | (0.07680–0.08862] | 7 | 2311 (10) | 4.5 | 8.2 |
| | (0.08864–0.10636] | 8 | 2312 (10) | 7.1 | 9.7 |
| | (0.10636–0.1366] | 9 | 2311 (10) | 7.5 | 11.9 |
| | (0.1366–0.8814] | Decile 10 | 2312 (10) | 15.5 | 20.0 |
| **LACE+** | [0–0.051] | No risk | 19,512 (85) | 3.6 | 1.8 |
| | (0.051–0.102] | Low risk | 2621 (11) | 10.8 | 7.1 |
| | (0.102–0.153] | Medium risk | 663 (3) | 18.3 | 12.1 |
| | (0.153–1] | High risk | 320 (1) | 20.3 | 19.6 |
| | [0.00355–0.00791] | Decile 1 | 2362 (10) | 1.9 | 0.7 |
| | (0.00792–0.00945] | 2 | 2263 (10) | 1.6 | 0.9 |
| | (0. 00945–0.01153] | 3 | 2314 (10) | 2.5 | 1.0 |
| | (0.01155–0.01373] | 4 | 2313 (10) | 2.6 | 1.2 |
| | (0.01373–0.01685] | 5 | 2306 (10) | 3.4 | 1.5 |
| | (0.01685–0.02115] | 6 | 2312 (10) | 4.2 | 1.9 |
| | (0.02115–0.02831] | 7 | 2313 (10) | 4.4 | 2.4 |
| | (0.02831–0.04147] | 8 | 2310 (10) | 6.8 | 3.4 |
| | (0.04151–0.06873] | 9 | 2313 (10) | 9.1 | 5.3 |
| | (0.06875–0.35077] | Decile 10 | 2310 (10) | 14.6 | 11.0 |
| **SQLape®** | [0–0.051] | No risk | 18,297 (80) | 3.5 | 1.8 |
| | (0.051–0.102] | Low risk | 4271 (18) | 10.2 | 6.9 |
| | (0.102–0.153] | Medium risk | 512 (2) | 18.3 | 12.2 |
| | (0.153–1] | High risk | 36 (0) | 33.3 | 17.9 |
| | [0.00257–0.00492] | Decile 1 | 4289 (18) | 2.0 | 0.5 |
| | (0.00496–0.00496] | 2 | 472 (2) | 1.5 | 0.5 |
| | (0. 00522–0.00905] | 3 | 2252 (10) | 2.0 | 0.7 |
| | (0.01002–0.01528] | 4 | 2290 (10) | 2.2 | 1.3 |
| | (0.01532–0.02188] | 5 | 2422 (10) | 2.8 | 1.9 |
| | (0.02202–0.02842] | 6 | 2210 (10) | 5.2 | 2.6 |
| | (0.02861–0.04063] | 7 | 2258 (10) | 4.9 | 3.5 |
| | (0.04110–0.05426] | 8 | 2328 (10) | 7.7 | 4.7 |
| | (0.05505–0.06087] | 9 | 2295 (10) | 7.2 | 6.0 |
| | (0.06146–0.18908] | Decile 10 | 2300 (10) | 15.5 | 9.3 |

(A) Risk intervals were rounded to the 5th decimal place.

(B) Deciles are based on the predicted probabilities not on the number of inpatients.

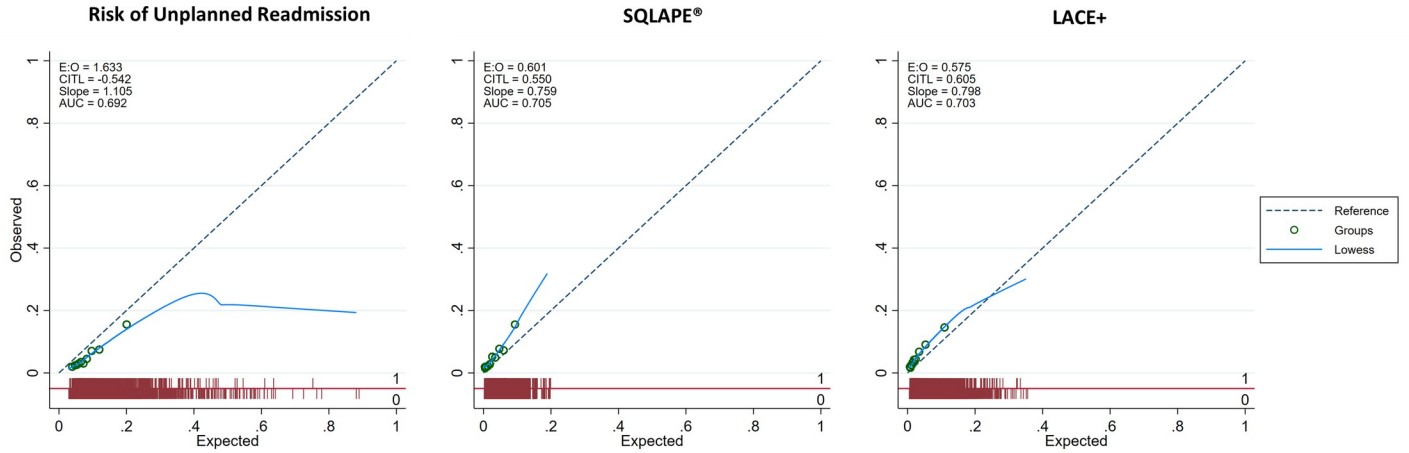

**Fig 2. Calibration plots Cohort A.** (A) Abbreviations: AUC–Area under the curve; CITL–Calibration-in-the-large; E:O–Expected: Observed. (B) Notification: Associated 95% CI were too narrow to be clearly displayed.

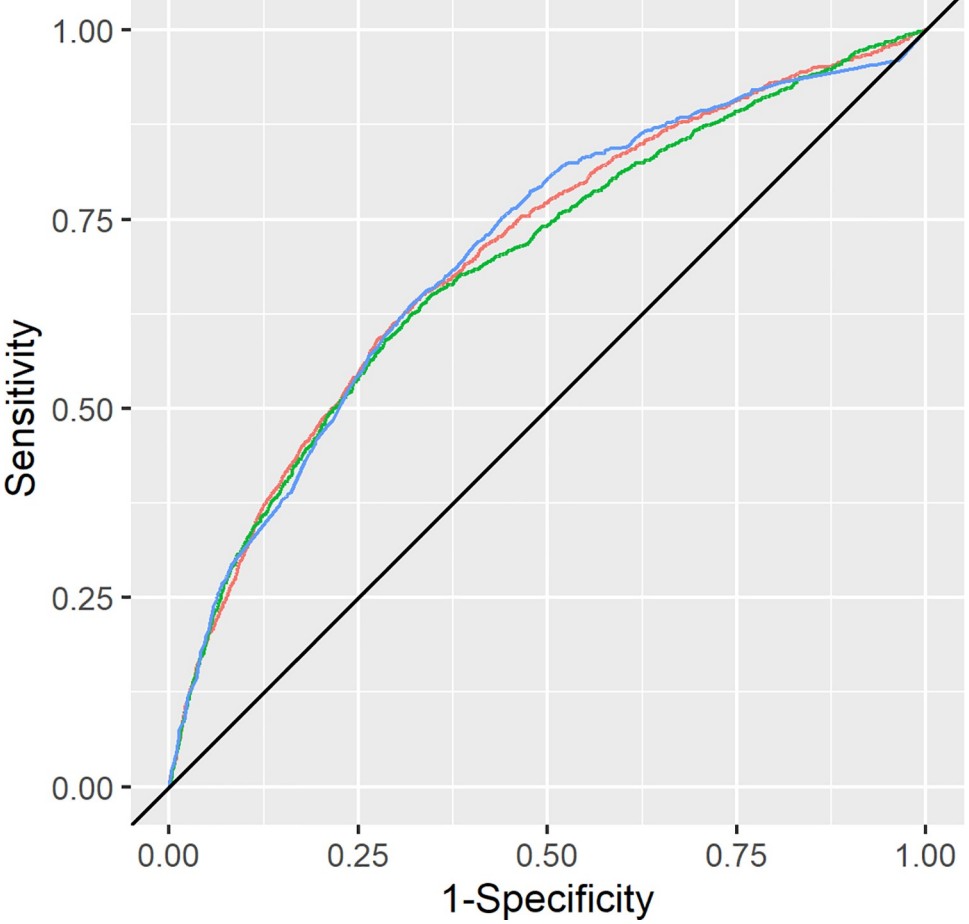

**Fig 3. Receiver operating characteristic curves.** (A) Red graph line = LACE+, Green = Epic model, Blue = SQLape®.

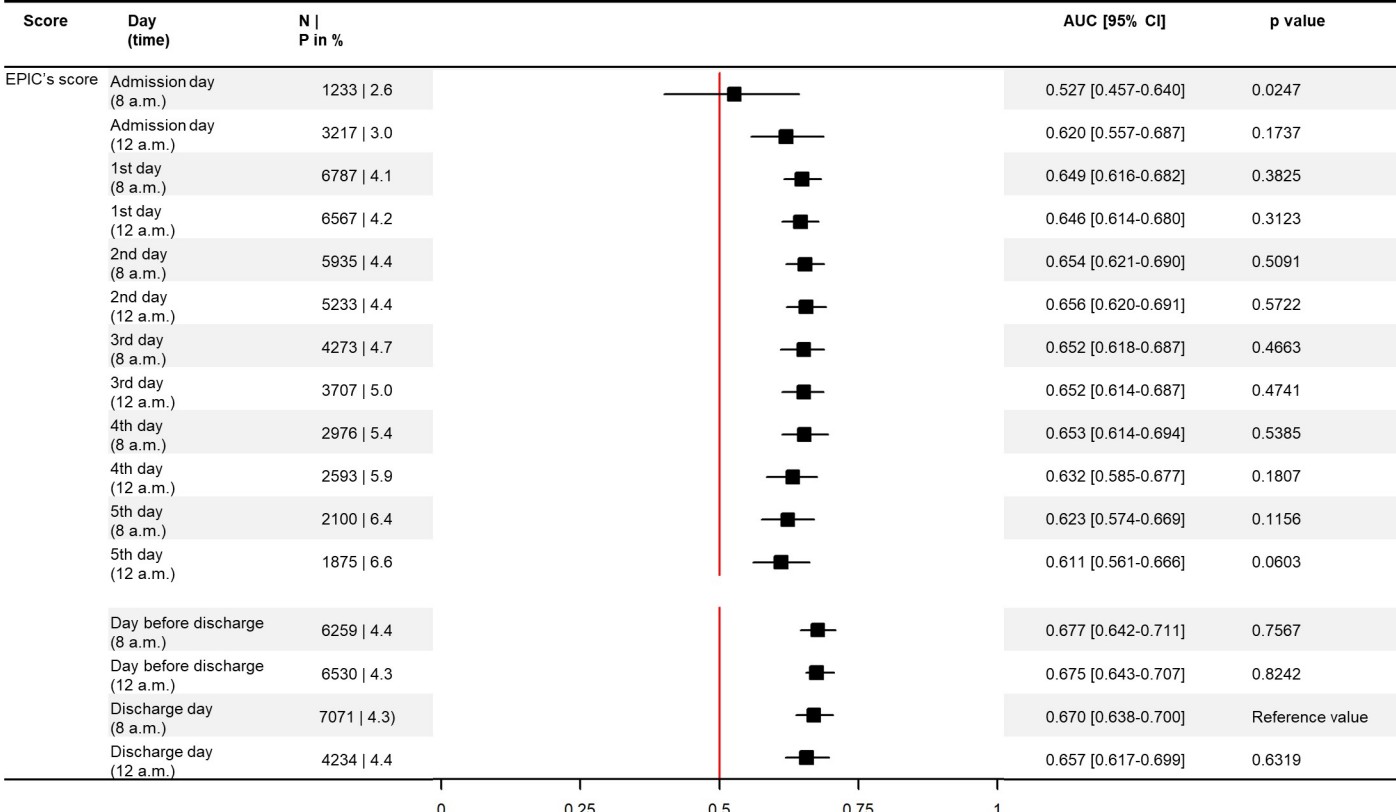

**Fig 4. Forest plot–Epic predictive ability (AUC) at different times throughout the hospital stay.** (A) Abbreviations: P = prevalence of the event of interest–unplanned readmissions.

account for approximately 10% of all readmissions [8]. This may have contributed to readmission rates being at the lower end (5.1%) compared to the derivation studies (ranging from 5.2 to 16.9%). In addition, patients who died after index hospital discharge were not excluded (e.g., by contacting each discharged patient 30 days after discharge). Both limitations may have led to over- or underestimations. Fourth, this study applied a different endpoint than the one originally investigated in the derivation study of the Epic and SQLape® model. As described earlier, the Epic and SQLape® models targeted the same basic endpoint (compared to LACE +) but used a more sophisticated definition. However, more sophisticated approaches may allow to overcome the lack of precision of the one used in the LACE+ derivation study (i.e., using unplanned readmissions as a proxy of potentially avoidable readmissions), they have not yet become standard in research studies and thus encumber benchmarking. Consequently, there is a chance of reduced performance for the Epic and SQLape® models. Last, it must be acknowledged that this study was conducted based on the assumption that if a specific condition, order, or test result was not documented in the medical records, it was absent/not prescribed/negative. This design is less powerful than a prospective study in which each variable would be collected and documented (positive and negative answers).

## Interpretation

In this study, EMR data were used to externally validate the Epic Risk of Unplanned Readmission model and to compare it with the LACE+ and SQLape® models. Until the date of submission, this was the first external study scientifically validating the Epic model as a predictor

of unplanned readmission within 30 days of hospital discharge. The principal findings can be summarized as follows. The performance measure Brier score indicates a superior, comparative overall performance of the SQLape® (for Cohort A, scores were 0.0484 for the Epic model, 0.0474 and 0.0473 for the LACE+ and the SQLape® models, respectively).

The Epic Risk of Unplanned Readmission model has poor discrimination and calibration to predict unplanned readmissions at a Swiss tertiary-care hospital. In comparison, LACE + and SQLape® yielded acceptable discrimination and calibration for the same study population (Cohort A). The discriminatory ability of the Epic model (AUC: 0.692) was at the lower end of the results provided by the developers (AUC ranged from 0.69 to 0.74). In terms of calibration, reference results were not available for the derivation study. The observed discrepancies can be explained by the differences between the study populations and outcomes. According to EPIC systems, the development sample had a much higher rate of readmissions (16.9 vs 5.1%), a higher sample size (203,500 vs. 23,116), and younger subjects (mean 48 vs. 51 years). Using current national survey data as a reference, it is most likely that variables describing the utilization of healthcare resources (particularly the number of ED visits and hospital discharges), as well as also practice patterns (e.g., imaging order rates) were diverging as well [44–47].

Similarly, the discriminatory ability of LACE+ (AUC: 0.703) proved to be lower than that in the original derivation study, where the AUC ranged from 0.749 to 0.757 [27]. Admittedly, this comparison has one limitation; that is, the 95% CI was computed including the CMG variable, which can only be calculated for hospitals in Canada. An AUC of 0.743 was reported for LACE+ without CMG (all patients were assigned 0 points for the CMG score). Following a general rule of thumb, LACE+ falls into a category of more discriminative models (0.7 ≤ AUC < 0.8; "acceptable") by marginally outperforming the Epic model [29]. Unfortunately, distribution and calibration data were reported only for the combined 30-day death or unplanned readmission outcome. The calibration graph of this study showed very similar rates (observed vs. expected) but demonstrated a tendency to overestimate at higher risk values. In the derivation study, these high-risk patients represented only approximately 1.6% of the validation set. Although there was no difference in the outcome and the exclusion criteria were fairly similar, the selected patient characteristics differed substantially. The original derivation study, compared to this study, had many older subjects (58 vs. 51 years) and significantly higher rates of ED visits (37.7% ≥ 1 ED visit the in previous 6 months vs. 11.4%) and hospitalizations (14.0% ≥ 1 urgent admission in the previous year vs. 1.7%). Additionally, 64.4% of patients included in the original derivation study were admitted urgently compared to 53.1% in this study. LACE+ has recently been investigated in various retrospective studies. However, samples were highly fragmented and ill-fitted for comparison [48–52].

The SQLape® model, which was developed based on a large Swiss development sample, demonstrated deviance to a lesser extent. The data shown in this study indicate that the SQLape® has an acceptable discriminatory ability. Compared to the original derivation study, the AUC was only slightly lower (AUC: 0.705 vs. 0.720), but the 95% CI was intersecting (0.690–0.720) [6]. In terms of calibration, no reference values were available. Although SQLape® was designed specifically to focus on readmissions that are potentially avoidable, model performance did not appear to have been affected tremendously in this study. Presumably, this can be attributed to overall few differences in methodology. Ultimately, in regard to the SQLape® model, this study is more like a temporal validation than an external validation study, i.e., including new individuals from the same institution but in a different time period [16].

For the assessment of discriminative ability, the NRI provides supplementary information that, however, needs to be interpreted in light of known limitations of the NRI [38, 53–55].

The main criticisms are that the NRI is highly sensitive to the number of risk categories and thresholds and is unstable when used to compare miscalibrated models. According to the net reclassification improvement (NRI), the LACE+ and Epic models yielded minimally better performance than the reference standard SQLape®. For LACE+, the NRI components (events and nonevents) indicate that LACE+ is slightly better at the detection of subjects with and without the outcome of interest. In regard to the Epic model, the results are inconsistent–a strong improvement in the detection of subjects with the outcome is indicated; in contrast, the detection of subjects without the outcome worsens.

In summary, the Epic Risk of Unplanned Readmission model performs similarly to its direct comparators (LACE+ and SQLape®) and other commonly used prediction models. As a broader comparison, the well-known LACE (the predecessor model of the LACE+), the HOSPITAL, and the PAR-Risk score, all validated in the Swiss population, have a C-statistic of 0.73 or less in external validation studies [18–21]. Compared to the promotional information of EPIC Systems, it must be noted that the performance did not meet expectations. Presumably, this can be explained by the differences in patient populations for model fit and external validation. Despite a rather comprehensive approach, the Epic model includes 27 variables from various domains and was outperformed by both comparators. For example, LACE+ and SQLape® do not consider medications but focus on admission characteristics (index admission type, existing comorbidities, health service utilization, etc.). In this regard, a recent study showed that prescription of corticosteroids and antidepressants was associated with a greater risk of unplanned readmission [56]. In conclusion, it may be assumed with caution that the Epic model´s relative complexity (compared to its comparators) has hampered its generalizability to different patient populations and settings–model updating is therefore warranted. An important advantage of the Epic model is that risk scores can be generated anytime throughout the hospital stay. Admittedly, the main caveat in this regard is its unstable predictive ability depending on day and time, as was pointed out during the subgroup analysis of discriminative ability.

## Implications for clinical practice

As the digital transformation has reached the healthcare sector, hospitals increasingly invest in IT systems and competencies to be able to turn immense volumes of data into actionable insights. For that matter, predictive analytics and machine learning have become some of the most discussed disruptive innovations in healthcare. In the context of unplanned hospital readmissions, the introduction of EHR systems has alleviated major limitations, such as the restricted accessibility of appropriate predictors and delayed reporting capabilities. Medical facilities adopting EPIC´s EHR system are advised to look at its "AI & Analytics" module, containing the Epic Risk of Unplanned Readmission model and others. Despite its underperformance in both respects (compared to investigated competitors and promotional information), for organizations switching from paper-based documentation or clinical data housed in multiple disconnected systems, the use of the Epic prediction model holds great potential to improve efficiency in service provision and patient outcomes. The model allows real-time identification of patients at high risk for unplanned hospital readmission. Incorporated clinically actionable variables, such as medications, that could be used to triage patients to different types of interventions are another noteworthy model feature. According to Leppin and colleagues, effective interventions are complex and seek to enhance patient capacity to reliably access and enact postdischarge transitional care. Their findings also suggest that providing comprehensive and context-sensitive support reduces the risk of hospital readmission within 30 days; the overall pooled relative risk of readmission was 0.82 (95% CI, 0.73–0.91; p<0.001)

[12]. In general, interventions included anywhere from 1 to 7 unique activities, including case management, patient education, medication intervention, and timely follow-ups.

In regard to the optimal moment to identify patients at highest risk (i.e., facilitating the best predictive performance), it may not be worth the loss of sufficient lead time to leverage additional data on disease progression and hospital complications. Speaking of trade-offs relevant to the application in routine care, a value- or "utility"- based decision is also indicated in regard to the threshold–with the costs of misclassification (e.g., expressed in terms of mortality and morbidity) on one side, and the allocation of resources (e.g. for subsequent detailed assessments or interventional measures) on the other. Since EPIC system´s predictive analytics platform update, as of then powered by cloud, updating EPIC developed models based on setting specific data prior to routine application is possible. Although this component is subject to charge, improvement in performance may warrant the expenditure of financial means.

## Implications for research

This study raises a number of opportunities for future research, both in terms of model validation and updating. First, it has become common practice to compare prediction models based on their performance on the day of discharge. However, given that most transitional care interventions need lead time, predictive risk scores should ideally provide information early enough during hospitalization [12]. Therefore, to contribute to clinicians´ choice of the best prediction model, validation studies should imbed adequate sample sizes that allow comprehensive subgroup analysis, i.e., provide information in regard to the predictive ability at different times throughout the hospital stay. Second, the Epic model showed reduced predictive performance; in particular, calibration was rather disappointing. Different sources may have distorted the calibration, including discrepancies in patient characteristics, outcome prevalence, and definition, and systematic differences in measurement errors. When a prediction model performs inadequately during external validation, it has been shown that the model can often be updated using data from the validation setting. Updating of regression-based algorithms may vary from simply changing the intercept (for differences in outcome frequency), adjusting the relative weights of the predictors (to represent setting specific associations of the predictors), to adding new predictors [16, 57]. Third, despite verified associations, only a few prediction tools for unplanned readmission include environmental and/or functional status [14, 58–61], as they are rarely readily available (e.g., data are often housed in multiple disconnected systems or paper-based systems). With the advancing adoption of commercial EHR systems and their instantaneous potential to provide clinically granular data from the entire course of hospitalization, factors such as living situation and nursing scores need further investigation. Fourth, a potential path to developing more comprehensive patient-risk models is machine learning, which has proven to be able to process extremely large numbers of input features and to be typically more predictive than standard logistic regression methods [62–65]. Last, interventional research is now needed to better understand the effects of risk prediction scores followed by available transitional care measures.

## Supporting information

**S1 Appendix. Anatomical Therapeutic Chemical Classification System (ATC)—subgroups.**
(DOCX)

**S2 Appendix. Laboratory components and corresponding reference ranges.**
(DOCX)

**S3 Appendix. Detailed description of prediction model variables.**
(DOCX)

**S4 Appendix. Cohort B–Brier scores.**
(DOCX)

**S5 Appendix. Calibration plots by risk group thresholds, Cohort A.**
(DOCX)

**S6 Appendix. Calibration plots–Cohort B.**
(DOCX)

**S7 Appendix. Reclassification tables–Cohort A.**
(DOCX)

## Acknowledgments

The authors would like to thank Johannes Rogger, clinical pharmacist and EPIC Willow Analyst, for the development of Swiss specific therapeutic subgroups of the Anatomical Therapeutic Chemical Classification System, based on the original US specifications and with regard to local regulations and classifications. Also, the authors would like to thank Dr. med. Pius Estermann (clinical coder) who was responsible for the mapping of ICD-10-CM codes onto ICD-10 codes of the German Modification (GM) version, Sebastian Zimmermann (laboratory analyst) for his critical review of laboratory input parameters and Madlene Michel (Head of Discharge Management) for her collaboration and valuable expertise. For statistical consultancy, and methodological advice, the authors thank Dr. Dirk Lehnick, Head Biostatistics & Methodology CTU-CS, and Prof. Dr. Konstantin Beck, former director of the CSS Institute for empirical Health Economics. Finally, the authors are very grateful to Roger Wicki for providing precious positivity and considerable technical support in regard to data curation, model deployment and maintenance.

## Author Contributions

**Conceptualization:** Aljoscha Benjamin Hwang, Guido Schuepfer, Stefan Boes.

**Data curation:** Aljoscha Benjamin Hwang, Mario Pietrini.

**Formal analysis:** Aljoscha Benjamin Hwang.

**Investigation:** Aljoscha Benjamin Hwang.

**Methodology:** Aljoscha Benjamin Hwang, Guido Schuepfer, Stefan Boes.

**Project administration:** Aljoscha Benjamin Hwang.

**Resources:** Aljoscha Benjamin Hwang, Guido Schuepfer, Mario Pietrini.

**Supervision:** Guido Schuepfer, Mario Pietrini, Stefan Boes.

**Validation:** Mario Pietrini, Stefan Boes.

**Visualization:** Aljoscha Benjamin Hwang.

**Writing – original draft:** Aljoscha Benjamin Hwang.

**Writing – review & editing:** Guido Schuepfer, Mario Pietrini, Stefan Boes.

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
