## [Decision Letter · Decision Letter 0]

22 Jun 2021

PONE-D-21-01674

External validation of EPIC’s Risk of Unplanned Readmission model, the LACE+ index and SQLape® as predictors of unplanned hospital readmissions: A monocentric, retrospective, diagnostic cohort study in Switzerland

PLOS ONE

Dear Dr. Hwang,

Thank you for submitting your manuscript to PLOS ONE. After careful consideration, we feel that it has merit but does not fully meet PLOS ONE’s publication criteria as it currently stands. Therefore, we invite you to submit a revised version of the manuscript that addresses the points raised during the review process.

Please answer the few questions from Reviewers and improve the English Languange before resubmitting the manuscript.

We look forward to receiving your revised manuscript.

Kind regards,

Michele Provenzano

Academic Editor

PLOS ONE

Journal Requirements:

Reviewers' comments:

Reviewer's Responses to Questions

**Comments to the Author**

1. Is the manuscript technically sound, and do the data support the conclusions?

Reviewer #1: Yes

Reviewer #2: Yes

2. Has the statistical analysis been performed appropriately and rigorously? 

Reviewer #1: Yes

Reviewer #2: Yes

3. Have the authors made all data underlying the findings in their manuscript fully available?

Reviewer #1: Yes

Reviewer #2: Yes

4. Is the manuscript presented in an intelligible fashion and written in standard English?

Reviewer #1: Yes

Reviewer #2: Yes

5. Review Comments to the Author

Reviewer #1: In my opinion this scientific research is technically valid and conclusions are supported by data and observation rigorously conducted.

The analyzed sample (23.116 records) and the length of study (from the 1st of January 2018 to the 31st of December 2019) have provided a statistically relevant set of data on a topic of great scientific and economic interest.

The data underlying the results presented in the study are appropriately descripted in the manuscript and available on request from the corresponding authors.

The manuscript is clear and comprehensible. I haven't found any typographical or grammatical errors.

Reviewer #2: This very interesting paper provides relevant results regarding the externally validation of the Epic Risk of Unplanned Readmission model as a predictor of unplanned hospital readmissions within 30 days and to compare its predictive ability with that of the LACE+ index and the SQLape® readmission algorithm. The study outcome was unplanned 30-day readmission to the same hospital. An unplanned readmission was defined as an urgent readmission, i.e., not scheduled in advance and requiring treatment within 12 hours. Methods are appropriate, results are clearly described and illustrated, as well as properly discussed. References are relevant and updated. Therefore, this paper requires minor corrections and can be very useful for Plos One readers, because it provides very interesting information within the current context of published studies. English language should be improved and revised.

6. PLOS authors have the option to publish the peer review history of their article (what does this mean?). If published, this will include your full peer review and any attached files.

Reviewer #1: **Yes: **Alessio Alò

Reviewer #2: No

---

## [Editor Report · Decision Letter 1]

27 Sep 2021

External validation of EPIC’s Risk of Unplanned Readmission model, the LACE+ index and SQLape® as predictors of unplanned hospital readmissions: A monocentric, retrospective, diagnostic cohort study in Switzerland

PONE-D-21-01674R1

Dear Dr. Hwang,

We’re pleased to inform you that your manuscript has been judged scientifically suitable for publication and will be formally accepted for publication once it meets all outstanding technical requirements.

Kind regards,

Michele Provenzano

Academic Editor

PLOS ONE
---

## [Editor Report · Acceptance letter]

3 Nov 2021

PONE-D-21-01674R1 

External validation of EPIC’s Risk of Unplanned Readmission model, the LACE+ index and SQLape as predictors of unplanned hospital readmissions: A monocentric, retrospective, diagnostic cohort study in Switzerland 

Dear Dr. Hwang:

I'm pleased to inform you that your manuscript has been deemed suitable for publication in PLOS ONE. Congratulations! Your manuscript is now with our production department. 

Kind regards, 

on behalf of

Dr. Michele Provenzano 

Academic Editor

PLOS ONE